# Recent Advances in Propylene-Based Elastomers Polymerized by Homogeneous Catalysts

**DOI:** 10.3390/polym16192717

**Published:** 2024-09-25

**Authors:** Chengkai Li, Guoqiang Fan, Gang Zheng, Rong Gao, Li Liu

**Affiliations:** Department of Polyethylene, SINOPEC (Beijing) Research Institute of Chemical Industry Co., Ltd., Beijing 100013, China; fangq.bjhy@sinopec.com (G.F.); zhenggang.bjhy@sinopec.com (G.Z.); gaor.bjhy@sinopec.com (R.G.); liul.bjhy@sinopec.com (L.L.)

**Keywords:** propylene-based elastomers, homogeneous catalysts, chain structures, physical properties

## Abstract

Propylene-based elastomers (PBEs) have received widespread attention and research in recent years due to their structural diversity and excellent properties, and are also an important area for leading chemical companies to compete for layout, but efficient synthesis of PBEs remains challenging. In this paper, we review the development of PBEs and categorize them into three types, grounded in their unique chain structures, including homopolymer propylene-based elastomers (hPBEs), random copolymer propylene-based elastomers (rPBEs), and block copolymer propylene-based elastomers (bPBEs). The successful synthesis of these diverse PBEs is largely credited to the relentless innovative advancements in homogeneous catalysts (metallocene catalysts, constrained geometry catalysts, and non-metallocene catalysts). Consequently, we summarize the catalytic performance of various homogeneous catalysts employed in PBE synthesis and delve into their effect on molecular weight, molecular weight distribution, and chain structures of the resulting PBEs. In the end, based on the current academic research and industrialization status of PBEs, an outlook on potential future research directions for PBEs is provided.

## 1. Introduction

Polyolefin materials have garnered immense attention ever since their inception, remaining a foremost choice among the most prevalent materials in contemporary applications [1,2,3,4,5]. Polyolefin thermoplastic elastomers (P-TPEs) seamlessly combine the processability of thermoplastics and the elasticity of rubber, thus broadening the scope of applications and enhancing the performance of traditional materials alike. These P-TPEs typically consist of hard and soft segments, with the hard phases serving as physical cross-links to provide the stress resistance and the soft phases providing superior elasticity [6,7,8,9,10]. Among the various P-TPEs, propylene-based elastomers (PBEs), with propylene as the primary monomer (>70% in most cases), excel in rheological, mechanical, optical, and processing characteristics, making them a prime choice for automobiles, household appliances, hygiene products, and packaging [11,12,13,14]. Furthermore, blending PBEs with other polymer materials can significantly enhance product performance, further broadening their utility. For instance, Fasihi et al. discovered that integration of a PBE into polypropylene (PP) bolstered its impact strength by a staggering 400% when incorporating 30 wt% of PBE [15]. A study by Gao et al. utilizing scanning electron microscopy (SEM) and dynamic mechanical analysis (DMA) techniques revealed that the compatibility between PP and PBE is superior to that between PP and polyolefin elastomers (POEs) [16]. Notably, as the PBE content increased, the blend’s crystallinity decreased, yet its melting temperature (*T*_m_) and crystallization temperature (*T*_c_) remained largely unchanged, resulting in enhanced mechanical properties. However, due to the intricate production process and high technical demands of PBEs, only a few of companies globally possess the capability to commercially manufacture PBEs, such as propylene-butene copolymer (Tafmer) from Mitsui Chemicals, Inc. (Tokyo, Japan), propylene-ethylene copolymer (Vistamaxx) from Exxon Mobil Corporation (Houston, TX, USA), and propylene-ethylene copolymer (Versify) from Dow Chemical Company (Midland, MI, USA).

Numerous parameters can significantly influence the properties of PBEs, among which, the molecular weight and molecular weight distribution of the polymers play a pivotal role in determining PBE properties; for example, high molecular weight atactic PP (*^a^*PP) exhibits elastomeric properties, while low molecular weight *^a^*PP is liquid or takes a viscous oil form. Furthermore, the microstructure of the polymer is another important factor in its material properties. Specially, isotactic PP (*^i^*PP), with regularly arranged stereocenters, is crystalline with notably high *T*_m_ (165 °C) and syndiotactic PP (*^s^*PP) is also crystalline, but the elastic modulus, impact strength, and crystallization rates vary, leading to distinct mechanical properties. In contrast, *^a^*PP is amorphous without *T*_m_, but with a low glass transition temperature (*T*_g_ = −12~0 °C), which shows distinct properties in comparison to both *^i^*PP and *^s^*PP (Figure 1). The current commercialization of PBEs is primarily centered on random copolymers of propylene and α-olefins. However, given the vast diversity in PP chain structures, there remain numerous high-performance PBEs awaiting further development, including homopolymer polypropylene-based elastomers (hPBEs) and block copolymer polypropylene-based elastomers (bPBEs). Each type of PBE possesses a distinctive chain structure, and the cornerstone lies in the difference in catalysts (Figure 1). Primarily, PBE synthesis hinges on the utilization of homogeneous catalysts, including metallocene catalysts, constrained geometry catalysts, and non-metallocene catalysts, featuring well-defined molecular structure, well-defined active site, and a diverse ligand, to achieve well-defined polymers, setting it apart from traditional Ziegler–Natta catalysts (Figure 2) [6,17,18,19,20,21,22,23,24,25]. Therefore, this paper reviews the catalytic activity and copolymerization ability of various homogeneous catalysts used for the synthesis of PBEs, as well as the effect of chain structure, molecular weight, molecular weight distribution of the resultant PBEs on the mechanical properties. Ultimately, perspectives on the potential development direction of novel PBEs with diverse compositions are offered.

## 2. Homopolymer Polypropylene-Based Elastomers

*^i^*PP is a crystalline material with notably high *T*_m_ (165 °C) and *^a^*PP is an amorphous material with low *T*_g_ (−12~0 °C), and unique stereoblock PP with TPE properties can be obtained by ingeniously combining these two chain structures [26,27,28,29]. In 1959, Natta et al. pioneered the synthesis of PP elastomers using the heterogeneous Ziegler–Natta catalyst and attributed the elastic behavior to the stereoblock chain structure with isotactic and atactic blocks distributed along the polymer chain. But the inhomogeneous polymer composition and low catalytic activity hindered the further development of this catalytic system in the field of PBEs [30,31]. Chien et al. utilized *rac*-[ethylidene(1-*η*^5^-tetramethylcyclopentadienyl)(l-*η*^5^-indenyl)]dichlorotitanium (**1**, Figure 3) for the polymerization of PP in 1990, which obtained PP with high-weight average molecular weights (*M*_w_ = 127 and 164 kg/mol), narrow molecular weight distributions (*Ð* = 1.9 and 1.7), as well as remarkable mechanical properties and elastic recovery (Run 1, Table 1) [32]. The TPE properties stemmed from the stereoblock chain structure of the PP, which likely results from the presence of both stereoselective and non-selective isomeric states of the active species. These isomeric states underwent constant changes during the polymerization process, thus alternating between isotactic and atactic blocks.

In 1995, Waymouth et al. designed and synthesized an unbridged bis(indenyl)zirconium dichloride-bearing phenyl substituent in the 2-position of the indenyl moiety (bis(2-phenylindeny)zirconium dichloride, **2**, Figure 3) for the production of stereoblock PBEs, since it existed in both *C*_2_-symmetric and *C*_2v_-symmetric conformations with similar energy during polymerization due to the ligand rotation [33]. Specifically, the *C*_2_-symmetric conformation catalyzed the formation of *^i^*PP, while the *C*_2v_-symmetric conformation catalyzed to obtain *^a^*PP, with the catalyst alternating between these two extreme conformations (Figure 4a). Ingeniously, the incorporation of the phenyl group on the indene moiety regulated the rotation rate of the ligand, allowing for both monomer insertion and the synthesis of stereoblock PP simultaneously (Figure 4b). Elevating the propylene pressure or reducing the polymerization temperature improved the catalytic activity, molecular weight, and isotacticity, thereby allowing for further fine-tuning of PBE properties (Runs 2–6, Table 1). Physical property tests indicated a tensile strength of 462 psi, an ultimate elongation of 1210% and a 50% tensile set when elongated to 300%. Subsequently, they delved further into the effects of substituents and metal centers on polymerization [34,35]. Under identical polymerization conditions, a catalyst (bis(2-(3,5-dimethylphenyl)indeny)zirconium dichloride, **3**, Figure 3) with methyl substitution on the benzene ring exhibited the lowest catalytic activity (63–87 kg/(mol·bar·h)), yielding PP of low molecular weight (*M*_w_ = 81–174 kg/mol) (Run 7, Table 1). In contrast, catalyst bis(2-(3,5-bis(trifluoromethyl)phenyl)indeny)zirconium dichloride, (**4**, Figure 3) with trifluoromethyl substitution on the benzene ring, displayed comparable activity to catalyst **2** and produced PP with relatively higher isotacticity ([mmmm] = 45–73%) (Run 8, Table 1). When the metal center was substituted with hafnium, the catalytic activity remained unchanged, but the isotacticity reduced ([mmmm] < 21%). These effects likely stem from the substituent’s influence on the rate of monomer insertion, the rate of interconversion among the isomers, and the equilibrium concentration of the isomeric forms in the steady state.

Rieger et al. successfully synthesized an array of *C*_1_-symmetric metallocene catalysts (*rac*-[1-(9-*η*^5^-fluorenyl)-2-(2-methylbenz[*e*]-1-*η*^5^-indenyl)ethane]zirconium dichloride, **5**, *rac*-[1-(9-*η*^5^-fluorenyl)-2-(2-methyltetrahydrobenz[*e*]-1-*η*^5^-indenyl)ethane]zirconium dichloride, **6**, *rac*-[1-(9-*η*^5^-fluorenyl)-2-(5,6-cyclopenta-2-methyl-1-*η*^5^-indenyl)ethane]zirconium dichloride, **7**, Figure 3), exhibiting remarkable catalytic activities up to 3.2 × 10^4^ kg/(mol·[*C*_3_]·h), and resulting in PP with high *M*_w_ up to 230 kg/mol and [mmmm] between 20–80%, which can be tailored by adjusting polymerization conditions (Runs 1–5, Table 2) [36]. For instance, increasing the temperature or reducing the propylene concentration for catalyst **7**, high isotactic semicrystalline thermoplastics were obtained, while the reverse conditions yielded excellent TPEs (Figure 5a,b). In 2003, Equistar Chemicals, LP, employed an indenoindolyl-based catalyst (bis-(5-phenyl-5,10-dihydroindeno [1,2-b]indolyl)zirconium dichloride, **8**, Figure 3) to obtain stereoblock PP with [mmmm] ranging from 20 mol% to 60 mol%, ideal for TPE applications (Run 6, Table 2) [37]. Moving forward to 2018, Rieger et al. conducted a comparative study on the impact of various *C*_1_-symmetric *ansa*-metallocene catalysts with a varied 2,5,6-indenyl substitution pattern on polymerization and PBE performances (*rac*-[1-(9-*η*^5^-fluorenyl)-2-(5,6-cyclopenta-2-methyl-1-*η*^5^-indenyl)ethane]hafnium dichloride, **9**, *rac*-[1-(9-*η*^5^-fluorenyl)-2-(2-methyl-1-*η*^5^-indenyl)ethane]hafnium dichloride, **10**, *rac*-[1-(9-*η*^5^-fluorenyl)-2-(2-methyl-5,10-dihydro-1*H*-5,10-[1,2]benzenocyclopenta[*b*]anthracene)ethane]hafnium dichloride, **11**, *rac*-[1-(9-*η*^5^-fluorenyl)-2-(5,6-cyclopenta-2-phenyl-1-*η*^5^-indenyl)ethane]hafnium dichloride, **12**, *rac*-[1-phenyl-1-(9-*η*^5^-fluorenyl)-2-(5,6-cyclopenta-2-methyl-1-*η*^5^-indenyl)ethane]hafnium dichloride, **13**, Figure 3) [38]. At 0 °C, the molecular weights exceeded 1000 kg/mol using the hafnium-based catalysts **9**–**13**, whereas the ones of zirconium-based catalyst **7** were lower (Runs 7–11, Table 2). Intriguingly, as the temperature increased, the *M*_n_ gradually converged. Notably, the high molecular weight PP catalyzed by the hafnium-based catalysts exhibited both superior mechanical properties and excellent elastic recovery (Figure 5c,d).

High molecular weight *^a^*PP also possesses TPE properties, which were successfully achieved through various catalytic systems. Resconi et al. obtained high molecular weight *^a^*PP (*M*_w_ up to 4 × 10^5^ g/mol) using dimethylsilanediylbis(9-fluorenyl))zirconium dichloride **14** and dimethylsilanediylbis(9-fluorenyl))zirconium dimethyl **15** as catalysts that exhibited non-sticky elastomeric properties, contrasting with the liquid or viscous oil form of low molecular weight PP (Figure 6, Run 12, Table 2) [39]. Bochmann et al. achieved the same goal using titanocene-based catalyst (trimethyl)pentamethylcyclopentadienyltitanium **16** with *M*_w_ up to 4 × 10^6^ g/mol and narrow polydispersity (Figure 6, Run 13, Table 2) [40]. In 2017, Chen et al. efficiently catalyzed propylene polymerization using [(*t*-Bu)_3_P=N]CpTiCl_2_ **17** and [(*t*-Bu)_3_P=N]CpTiMe_2_
**18**, obtaining high molecular weight *^a^*PP elastomers (*M*_w_ ranging from 126 to 442 kg/mol) with narrow molecular weight distribution (*Ð* = 1.7~1.9) and remarkable catalytic activity (up to 650 kg/(mol·bar·h)) (Figure 6, Runs 9–11, Table 1) [41]. Characterization revealed that the obtained polymer has no *T*_m_ and is a colorless and transparent elastomer.

**Table 1 polymers-16-02717-t001:** Selected polymerization results by catalysts **1**–**4** and **17**–**22**.

Run	Catalyst	*T*_p_ (°C)	Pressure (bar)	Activity (kg/(mol·bar·h))	*M*_w_ (kg/mol)	*Ð*	[mmmm] (%)	Ref.
1	1	50	1.5	250	127	1.9	n.d.	[32]
2	2	0	1.3	208	213	1.5	11.6	[33]
3		0	6.1	280	604	1.8	17.4	
4		−25	1	1100	330	2.2	60	
5		25	2.4	213	203	3.2	22	
6		45	1	190	24	2.8	52	
7	3	25	2.4	63	81	2.5	15	[34]
8	4	25	2.4	208	243	3.2	51	[34]
9	17	20	5	12	442	1.7	n.d.	[41]
10		40	5	650	206	1.9	n.d.	
11	18	30	1	250	126	1.8	n.d.	[41]
12	19	60	2	2600	470	2.2	n.d.	[42]
13	20	30	2	11,790	1512	2.6	n.d.	[42]
14	21	60	2	1700	370	2.3	n.d.	[42]
15	22	60	2	100	96	4.5	n.d.	[42]

**Table 2 polymers-16-02717-t002:** Selected polymerization results by catalysts **5**–**16**.

Run	Catalyst	*T*_p_ (°C)	[C_3_] (mol/L)	Activity (kg/(mol·[C_3_]·h))	*M*_w_ (kg/mol)	*Ð*	[mmmm] (%)	Ref.
1	5	70	1.1	12,540	32.4	3.04	66.3	[36]
2	6	70	1.1	8870	19.9	1.98	78.6	[36]
3	7	30	1.1	2880	71.0	1.88	47.8	[36]
4		30	6.1	2570	230.0	1.91	24.9	
5		70	1.1	32,020	24.8	2.05	64.0	
6	8	25	bulk	1115 kg/mol	85.9	6.1	26	[37]
7	9	0	2.14	1100	2860	2.2	24	[38]
8	10	0	2.14	1200	2730	2.1	15	[38]
9	11	0	2.14	940	2880	2.4	11	[38]
10	12	0	2.14	1400	1650	1.5	74	[38]
11	13	0	2.14	1300	1680	1.4	3	[38]
12	14	50	10.56	1060	440	n.d.	n.d.	[39]
13	16	-45	bulk	5250 kg/(mol·h)	3966	2.0	n.d.	[40]

Most recently, in 2022, O’Hare et al. discovered that permethylindenyl-phenoxy titanium complexes (**19**–**22**, Figure 6) are exceptionally efficient for propylene polymerization with high activities up to 1.2 × 10^4^ kg/(mol·bar·h) and ultra-high molecular weight (UHMW, *M*_w_ up to 1.5 × 10^3^ kg/mol) (Runs 12–15, Table 1) [42]. UHMW *^a^*PP was transparent and showed a transmittance of 85% in the visible light region (Figure 7a). The mechanical properties analysis indicated that the ultimate tensile strength was 1.08 MPa at a strain of 184% and elongation at break was higher than 1900% and the hysteresis testing was extended to 120% and showed an excellent elastic recovery, which may be due to the physical crosslinking made by the ultra-high molecular weights, thus limiting the irreversible deformation by reducing chain mobility (Figure 7b,c). The performances were comparable to commercial TPEs such as the Kraton^TM^ styrene–butadiene–styrene block copolymers.

In 2023, Hainan Beiouyi Technology Co., Ltd. (Haikou, China) filed a patent for PBE, a process wherein they synthesized graft copolymers by employing atactic amorphous PP as the backbone, serving as a rubber phase, and medium tacticity PP as the side chains, constituting the plastic phase [43]. These plastic phases, randomly interspersed along the main chain, form reversible physical cross-links that penetrated into the rubber phase, ultimately imparting TPE characteristics. Medium tacticity PP was firstly obtained using *rac*-ethylenebis(indenyl)zirconocene dichloride with [mmmm] between 50–65%. The chain-end double-bond content exceeded 85%, which facilitated subsequent polymerization. Next, Me_2_Si(9-Flu)_2_ZrCl_2_ or *meso*-C_2_H_4_(Ind)_2_ZrCl_2_ was employed to catalyze the copolymerization of medium tacticity PP with propylene, resulting in graft copolymers with 4–15% medium tacticity PP incorporation. Mechanical property tests revealed elongation at break exceeding 750%, tensile strength greater than 8 MPa, and no yielding behavior.

## 3. Random Copolymer Propylene-Based Elastomers

Current commercial PBEs are primarily random copolymers of propylene and various α-olefins. For example, in 1995, Mitsui Chemicals, Inc. unveiled a series of *ansa*-metallocenes with diverse substituents, using *rac*-dimethylsilyenebis(2-methyl-4-propylindenyl)zirconium dichloride (**23**, Figure 8) as an illustrative example, to catalyze the copolymerization of propylene and 1-butene for the production of PBEs, exhibiting high comonomer incorporation and catalytic activities reaching up to 2.1 × 10^4^ kg/(mol·bar·h) and the resulting PBEs possessed low *T*_m_, remarkable heat resistance, as well as transparency, making them suitable for use as sealing film with stable heat sealing properties (Run 1, Table 3) [44]. Subsequently, in 2000, different aromatic substituents were introduced at the 4-position of the indenyl group by Basell polypropylen GmbH (*rac*-dimethylsilyene-bis(2-methyl-4-phenylindenyl)zirconium dichloride, **24** as an example, Figure 8) and the copolymerization of propylene (>85 wt%) with various α-olefins (ethylene, 1-butene, 1-hexene, and 4-methyl-1-pentene) were investigated, in which ethylene as comonomer exhibited the highest catalytic activity, enabling production of high molecular weight copolymers (*M*_w_ > 4 × 10^2^ kg/mol) with high viscosity number [45].

The introduction of diverse substituents at varying positions within the cyclopentadienyl and fluorenyl groups profoundly impacts the microstructure of PP, obtaining syndiotactic, isotactic, hemiisotactic PP, etc., which are crucial category for synthesis of PBEs. Notably, Mitsui Chemicals, Inc.’s patent in 2008 utilized this series of catalysts to catalyze the copolymerization of propylene and 1-butene (exemplified in dimethylmethylene (3-*tert*-butyl-5-methylcyclopentadienyl)(3,6-di-*tert*-butylfluorenyl)zirconium dichloride, **25**, Figure 8), showing high isotacticity ([mm] > 90%) and low 2,1-insertion [46]. The resulting PBEs exhibited outstanding flexibility, impact resistance, heat resistance, low-temperature heat sealability, and wide heat-sealing temperatures.

Exxon Mobil Corporation successfully utilized hafnium-based catalyst dimethylsilyl bis(indenyl)hafnium dimethyl, **26** in catalyzing the random copolymerization of propylene and ethylene for the preparation of PBEs, and achieved a balance between *T*_m_, elasticity, tensile strength, and flexural modulus by adjusting the amount of ethylene insertion to adjust the degree of crystallinity and thus the mechanical properties [47]. In an intriguing finding, Quijada et al. discovered that when the metal central to the catalyst was substituted with zirconium (dimethylsilyl bis(indenyl)zirconium dichloride, **27**, Figure 8), the copolymerization activities of propylene with 1-hexene or 1-octadecene actually increased with the addition of the comonomers, which could be attributed to fact that the comonomers reactivated the dormant species formed after 2,1-insertion, a phenomenon referred to as the “co-monomer effect” (Figure 9a) [48]. The stress–strain curves obtained for each copolymer showed that as the comonomer incorporation increased, there was a decrease in the stress, indicating a decrease in the crystallinity (Figure 9b).

In 2006, Exxon Mobil Corporation further employed catalysts *rac*-dimethylsilyene-bis(5,6,7,8-tetrahydro-2,5,5,8,8-pentamethyl-benz[f]indenyl)hafnium dimethyl, **28**, *rac*-diphenylsilyene-bis(5,6,7,8-tetrahydro-2,5,5,8,8-pentamethyl-benz[f]indenyl)hafnium dimethyl, **29**, *rac*-trimethylenesilyene-bis(5,6,7,8-tetrahydro-2,5,5,8,8-pentamethyl-benz[f]indenyl)hafnium dimethyl, **30**, *rac*-diphenylsilyene-bis(5,6,7,8-tetrahydro-5,5,8,8-tetramethyl-benz[f]indenyl)hafnium dimethyl, **31**, and *rac*-diphenylsilyene-bis(5,6,7,8-tetrahydro-2,5,5,8,8-pentamethyl-benz[f]indenyl)hafnium dimethyl, **32** (Figure 8) for both the homopolymerization of propylene and the copolymerization of propylene and ethylene, leading to the production of PBEs [49]. Differential scanning calorimetry (DSC) analysis revealed that the resulting PBEs possessed suitable crystallinity, coupled with exceptional softness, while still maintaining superior tensile strength and elasticity.

A thorough comprehension of the relationship between chain structures and properties enables the precise customization of PBEs to achieve the desired physical properties. In 2007, Rosa et al. employed four metallocene catalysts (**33**–**36**, Figure 8) to catalyze the copolymerization of propylene with ethylene, 1-butene, or 1-hexene, respectively, aiming to produce PBEs of varying compositions and delved into the influence of different compositions and chain structures on the crystallinity and mechanical properties of PBEs (Runs 2–18, Table 3) [50]. Among them, the copolymer of propylene and 1-butene exhibited the highest crystallinity, and increasing the amounts of 1-butene had a minor impact on the crystallinity, while ethylene or 1-hexene as comonomer led to a decrease in crystallinity as the concentration of comonomers increased, probably due to the fact that 1-butene units produce a disturbance of the crystallization of *^i^*PP lower than that produced by ethylene and 1-hexene units, despite the bigger size of 1-butene compared to ethylene, and even lower than that of stereodefects, which was mirrored in the mechanical properties, where the Young’s modulus was almost unchanged with increasing 1-butene concentration, while increasing the ethylene or 1-hexene concentration resulted in a reduction in Young’s modulus, which can improve the performance of these polymers by fine-tuning the concentrations of ethylene or 1-hexene, enabling their application as thermoplastics and TPEs.

In 2021, the China National Petroleum Corporation (CNPC) applied for a patent on the preparation of PBEs (**37** as example, Figure 8), in which the propylene content was between 80 and 95 mol%, and ethylene and 1-hexene were used as comonomers (Run 19, Table 3) [51]. Notably, the properties of PBEs can be regulated by adjusting the comonomer content and the substituents of the catalysts. The PBEs obtained exhibited high elongation at break and notably enhanced low-temperature impact strength, leading to exceptional impact resistance at lower temperatures.

Beyond metallocene catalysts, Dow Chemical Company employed dimethyl(pyridylamido) hafnium catalysts to synthesize the high molecular weight PP and propylene-ethylene copolymers (**38**, Figure 10a), where abundant ligands can modulate the catalytic properties [52]. The PBEs catalyzed by this catalytic system possessed some different properties compared to those obtained from metallocene catalysts, such as higher *T*_g_ at the same crystallinity and lower *T*_m_, which implied superior creep resistance and processability. In 2011, they prepared polyolefin elastomer compositions by blending PBEs obtained from this catalytic system with ethylene-α-olefin copolymers, which possessed low adhesion characteristics [53]. This blend not only facilitated the industrial production of PBEs but also yielded compositions with remarkable mechanical properties, low haze, high gloss, and other advantageous features.

**Table 3 polymers-16-02717-t003:** Selected polymerization results by catalysts **23** and **33**–**38**.

Run	Catalyst	Comonomer	[Comonomer] (mol/L)	*T*_p_ (°C)	Pressure (bar)	Activity	*M*_w_ (kg/mol)	*Ð*	*T_m_*	Ref.
1	23	1-butene	0.97	70	7	21,300 kg/(mol·bar·h)	n.d.	2.05	n.d.	[44]
2	33	ethylene	4.0	60–70	bulk	n.d.	292	2.1	130	[50]
3			7.4	60–70	bulk	n.d.	288	2.1	115	[50]
4	36		13.1	60–70	bulk	n.d.	193	2.0	55	[50]
5	33	1-butene	4.3	60–70	bulk	n.d.	228	2.1	137	[50]
6			8.2	60–70	bulk	n.d.	178	2.0	125	[50]
7	34		1.6	60–70	bulk	n.d.	225	2.0	139	[50]
8			2.8	60–70	bulk	n.d.	251	2.0	134	[50]
9			6.0	60–70	bulk	n.d.	229	2.0	123	[50]
10	35		1.3	60–70	bulk	n.d.	173	2.1	135	[50]
11			4.6	60–70	bulk	n.d.	176	2.0	125	[50]
12			8.2	60–70	bulk	n.d.	177	2.0	115	[50]
13	36		1.4	60–70	bulk	n.d.	214	2.1	128	[50]
14			2.2	60–70	bulk	n.d.	215	2.0	124	[50]
15			6.4	60–70	bulk	n.d.	214	2.0	114	[50]
16	33	1-hexene	1.2	60–70	bulk	n.d.	700	2.3	139	[50]
17			4.2	60–70	bulk	n.d.	292	2.0	109	[50]
18			11.2	60–70	bulk	n.d.	266	1.9	70	[50]
19	37	ethylene 1-hexene	1	90	19 mol/L	2490 kg/(mol·h))	n.d.	n.d.	117	[51]
20	38	3,3-dimethyl-3-sila-1,5-hexadiene	2.5 × 10^−4^	25	1	2940 kg/(mol·bar·h)	252	2.10	156.8	[54]
21			2 × 10^−3^	25	1	670 kg/(mol·bar·h)	726	2.13	100	
22			3 × 10^−3^	25	1	300 kg/(mol·bar·h)	700	1.56	n.d.	
23		5-(N,N-diisopropylamino)-1-pentene	1.9 × 10^−2^	25	1	1080 kg/(mol·bar·h)	156	2.1	129.7	[55]
24		5-(N,N-diphenylamino)-1-pentene	1.9 × 10^−2^	25	1	6580 kg/(mol·bar·h)	185	2.4	147.1	
25		1-octene	0.15	25	1	50,000 kg/(mol·bar·h)	519	1.4	46	[56]
26		1-dodecane	0.15	25	1	51,000 kg/(mol·bar·h)	960	1.4	33	
27		1-hexadecane	0.15	25	1	52,000 kg/(mol·bar·h)	1416	1.5	34	
28		1-eicosene	0.15	25	1	57,000 kg/(mol·bar·h)	1678	1.4	80	

In 2015, Li et al. catalyzed the copolymerization of propylene with asymmetric Si-containing α,ω-diolefins using the same catalytic system (Figure 10b), with excellent catalytic activity (up to 2.94 × 10^3^ kg/(mol·bar·h)), high comonomer incorporation (up to 25.3%), and high molecular weights of copolymers (*M*_w_ up to 7.26 × 10^2^ kg/mol), and crucially no cross-linking reaction occurred (Runs 20–22, Table 3) [54]. The *T*_m_ and crystallinity of the copolymers decreased with the increase of the comonomer concentration, enabling the fine-tuning of copolymer properties to obtain a series of materials ranging from thermoplastic to highly flexible to elastomeric (Figure 11a). Cycle tensile tests indicated that the elastic recovery is impressively high, above 80% over the whole range of deformations (Figure 11b). Furthermore, this α,ω-diolefins monomer can incorporate silicon atoms into the polymer chain, which is very advantageous for the development of new polyolefin materials.

In 2019, Pan et al. achieved the direct copolymerization of propylene with various amino-functionalized α-olefins (AO, N(pentenyl)*^i^*Pr_2_, N(pentenyl)Ph_2_) using catalyst **38** by introducing bulky groups to shield the polar atom to protect the highly oxophilic catalytic center from becoming poisoned (Figure 10c, Runs 23 and 24, Table 3), in which high catalytic activity (up to 7.44 × 10^3^ kg/(mol·bar·h)) was achieved for the 4-(7-octen-1-yl)-*N*,*N*-diphenylaniline comonomer [55]. The functional copolymers exhibited high molecular weight (*M*_w_ up to 5.9 × 10^2^ kg/mol), remarkable comonomer incorporation (up to 11.6 mol%), and high isotacticity ([mmmm] > 99%), resulting in a high *T*_m_ of 158 °C and high thermal stability, evidenced by a thermal decomposition temperature of approximately 450 °C (Figure 12a). Furthermore, the elastomer properties of these copolymers could be fine-tuned by adjusting the comonomer incorporation, achieving an elongation at break of up to 774% alongside a tensile strength of 22 MPa (Figure 12b). Advancing the aqueous contact angle measurement indicated that comonomer incorporation significantly modified the surface properties as the water contact angle decreased systemically from 104.8º of *^i^*PP to 82.1º of copolymer containing 7.8 mol % of comonomer (Figure 12c).

Wang et al., in 2020, investigated the copolymerization of propylene with various higher α-olefins by **38**, including 1-octene, 1-dodecene, 1-hexadecene, and 1-eicosene with high catalytic activity (up to 5.7 × 10^4^ kg/(mol·bar·h)) even at high comonomer incorporation (up to 18.2 mol%) and the molecular weights (*M*_W_, ranging from 300–1700 kg/mol) of the resultant copolymers increased gradually with the increase of comonomer content, which is different from the metallocene-catalyzed copolymerization of propylene with α-olefins (Figure 10d, Runs 25–28, Table 3) [56]. When the comonomer content reached above 12 mol%, PBEs were obtained with high ductility and excellent elastic recovery was achieved with about 20 mol% comonomer incorporation (Figure 13a,b), which could be attributed to the presence of small crystalline anchors due to the comonomer incorporation and the chain entanglement density in the amorphous phase (Figure 13c).

## 4. Block Copolymer Propylene-Based Elastomers

*^a^*PP, renowned for its low glass transition temperature, is an amorphous polymer that perfectly acts as soft segment within TPEs [57,58,59,60]. However, a limitation arises from traditional Ziegler–Natta catalysts, which are restricted to producing low molecular weight *^a^*PP that are less favorable for TPEs [42]. In contrast, homogeneous catalysts excel in catalyzing propylene polymerization, resulting in high molecular weight *^a^*PP that is ideal for the soft segments of TPEs. In 2015, Sita et al. employed a hafnocene-based catalyst **39** to catalyze the sequential cyclic/linear/cyclic living coordination polymerization of 1,6-heptadiene (HPD), propene, and HPD, respectively, yielding poly(1,3-methylenecyclohexane)-*b*-atactic polypropene-*b*-poly(1,3-methylenecyclohexane) (PMCH-*b*-*^a^*PP-*b*-PMCH) polyolefin triblock copolymer (Figure 14a, Run 1, Table 4), which PMCH served as hard segments due to its high *T*_g_ (72 °C) and *^a^*PP as soft segments (Figure 14b) [61]. Triblock copolymers exhibited high molecular weights (*M*_w_ = 1.8 × 10^2^ to 4.0 × 10^2^ kg/mol) and narrow molecular weight distributions (*Ð* = 1.03~1.18). By varying the weight fraction of hard segments, the material properties were tailored, achieving an elongation at break of up to 2773%, a tensile strength of up to 20.3 MPa, and elastic recovery as high as 94% after several of stress–strain cycles (Figure 14c).

In 2019, Shiono et al. utilized the (*tert*-butyl((2,7-di-*tert*-butyl-9*H*-fluoren-9-yl)dimethylsilyl)amino)dimethyltitanium catalyst **40** for the synthesis of A-B-A-type triblock copolymers (Figure 15a), in which the poly(norbornene) with high *T*_g_ (approximately 400 °C), served as the hard segment, while *^a^*PP, a random copolymer of ethylene and propylene or ethylene and 1-hexene, acted as the soft segment [62]. The triblock copolymers exhibited exceptional toughness and elastic recovery with a high molecular weight (*M*_w_ = 14–340 kg/mol). The hydroxyl can be incorporated by using 10-undecen-1-ol as a comonomer and the incorporation of hydroxyl groups not only optimized the surface properties but also bolstered the mechanical properties (Figure 15b, Run 2, Table 4) [62], which greatly improved the performance of the random copolymer, comprising norbornene and 1-octene as the hard segment material, which had been previously reported by the authors [63].

Not only are metallocene catalysts capable of catalyzing the living polymerization of propylene to produce block PBEs, but also non-metallocene catalysts are equally effective in achieving this purpose. In 2006, Coats et al. achieved the synthesis of triblock copolymers syndiotactic polypropylene-*b*-poly(ethylene-*co*-propylene)-*b*-syndiotactic polypropylene (*^s^*PP-*b*-PEP-*b*-*^s^*PP) by living, stereoselective insertion polymerization catalysts bis(N-(3,5-di-*t*-butylsalicylidene)pentafluoroaniline)titanium dichloride **41** (Figure 16a), wherein the *^s^*PP possessed high syndiotacticity ([rrrr] = 96%) and *T*_m_ ranged from 130 to 146 °C, which was suitable as the hard segment (Run 3, Table 4) [64]. The elongation at break of the triblock copolymer could reach 550% with better than 90% elastic recovery, which are comparable with or better than commercial polystyrene-*b*-poly(ethylene-co-butylene)-*b*-polystyrene (SEBS) triblock copolymers with cylindrical polystyrene microstructures. Furthermore, the authors employed a chiral nickel catalyst **42** to obtain *^i^*PP at -60 °C, and increased the temperature to 0 °C, *^a^*PP was produced, so that by regulating the polymerization temperature, *^i^*PP-*b*-*^r^*PP-*b*-*^i^*PP triblock copolymer TPEs can be obtained (Figure 16b, Run 4, Table 4), with the elongation at break of more than 1700%, and this methodology was further extended to synthesize *^i^*PP-*b*-*^r^*PP-*b*-*^i^*PP-*b*-*^r^*PP-*b*-*^i^*PP pentablock TPEs, exhibiting improved mechanical properties with elongation at break surpassing 2400%. Subsequently, the authors discovered that by modifying the substituents on the bis(phenoxyketimine) titanium catalyst bis(N-(3,5-dimethylsalicylidene)pentafluoroaniline)titanium dichloride **43**, living polymerization of the propylene was achieved with high isotacticity ([mmmm] = 73%) and *^i^*PP-*b*-PEP-*b*-*^i^*PP triblock copolymers were synthesized for the first time using the sequential monomer addition method (Figure 16c, Run 5, Table 4) [65]. Penta- and hepta-block copolymers could also be synthesized in the same way, and the obtained copolymers exhibited elastomeric properties, with triblock copolymer TPEs having an elongation at break up to 1000% and elastic recovery up to 80%.

**Table 4 polymers-16-02717-t004:** Selected polymerization results by catalysts **39**–**43**.

Run	Catalyst	*T*_p_ (°C)	*M*_w_ (kg/mol)	*Ð*	*T_m_* (°C)	Ref.
1	39	−15	403	1.18	n.d.	[61]
2	40	30/20	335	1.11	n.d.	[62]
3	41	0	358	1.20	36, 134	[64]
4	42	0/−60	124	1.14	130	[64]
5	43	0	306	1.30	95	[65]

Unlike the tri-block PBEs with well-defined structures, in 2016, LG Chem, Ltd. employed metallocene catalysts with different substitutes **44**–**47** to catalyze the copolymerization of propylene and ethylene with high catalytic activity (up to 834.3 kg/g), which resulted in PBEs containing an abundance of both propylene–propylene and ethylene–ethylene sequences (Figure 17) [66]. Compared with commercial PBEs, they exhibited higher density at the similar ethylene content and superior mechanical properties (elongation at break, flexural modulus, and tear strength, etc.).

## 5. Conclusions and Outlook

The past few decades have witnessed a remarkable evolution in the polyolefin industry, driven primarily by the relentless innovation of catalysts, enabling the manufacture of products with unparalleled performance. In this review, we overviewed a diverse array of catalysts employed for the synthesis of PBEs, covering data from both academia and industry, and explored the influence of catalysts on chain structures and physical properties. Although numerous homogeneous catalysts and chain structures of PBEs have been developed, most of the polymerization conditions differ from those of industrial polyolefin production. In view of the growing importance of PBEs in industry, continued optimization of catalysts and polymerization conditions will significantly enhance the potential and competitiveness of these catalysts for industrial applications.

Prospectively, the future evolution of PBEs should prioritize the following directions: firstly, the development of high-performance, cost-effective homogeneous catalysts with high-temperature stability, high catalytic activity, superior hydrogen-tuning sensitivity, and exceptional copolymerization capabilities that are more suitable for use in industrial polymerization plants. Secondly, the introduction of functional polar comonomers to enhance the performance of PBEs, such as printability, dyeability, adhesion, polymer miscibility and rheology, thus broadening their application fields. Thirdly, the development of mild and efficient post-polymerization modification to create PBEs with novel microstructures. Additionally, the pursuit of renewable monomers as comonomers would inject new vitality into the PBE industry. Lastly, promoting the research and development of closed-loop recyclable PBE materials to advance the industry’s sustainability efforts.

## Figures and Tables

**Figure 1 polymers-16-02717-f001:**
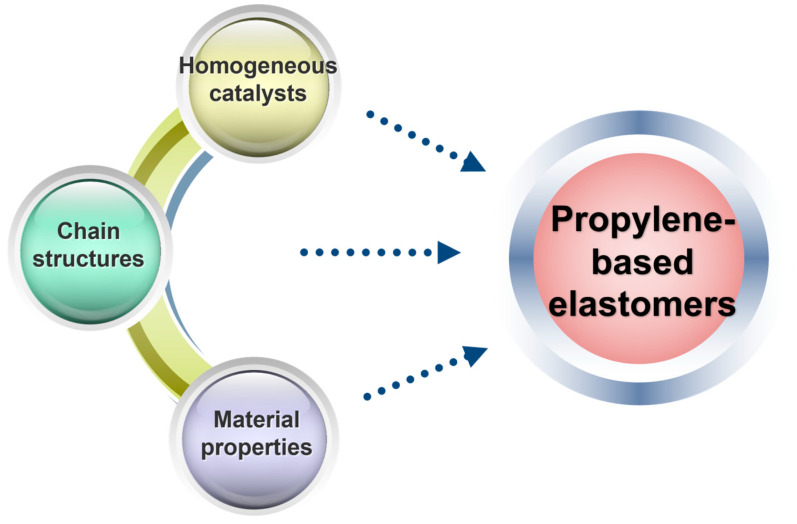
Key elements for the PBEs.

**Figure 2 polymers-16-02717-f002:**
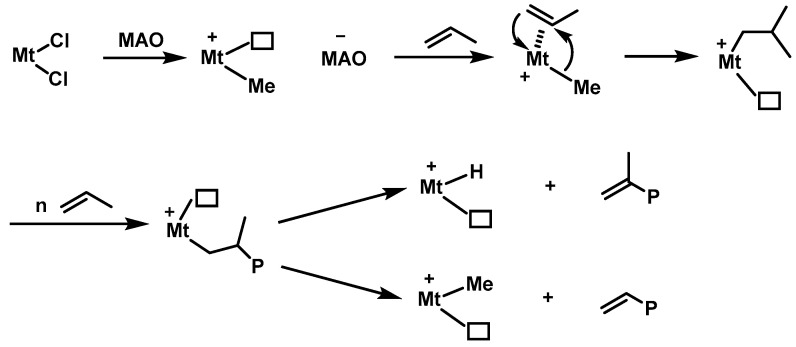
The polymerization mechanism of propylene by homogeneous catalysts (taking the polymerization of propylene as an example and some other chain transfer processes were not described for clarity).

**Figure 3 polymers-16-02717-f003:**
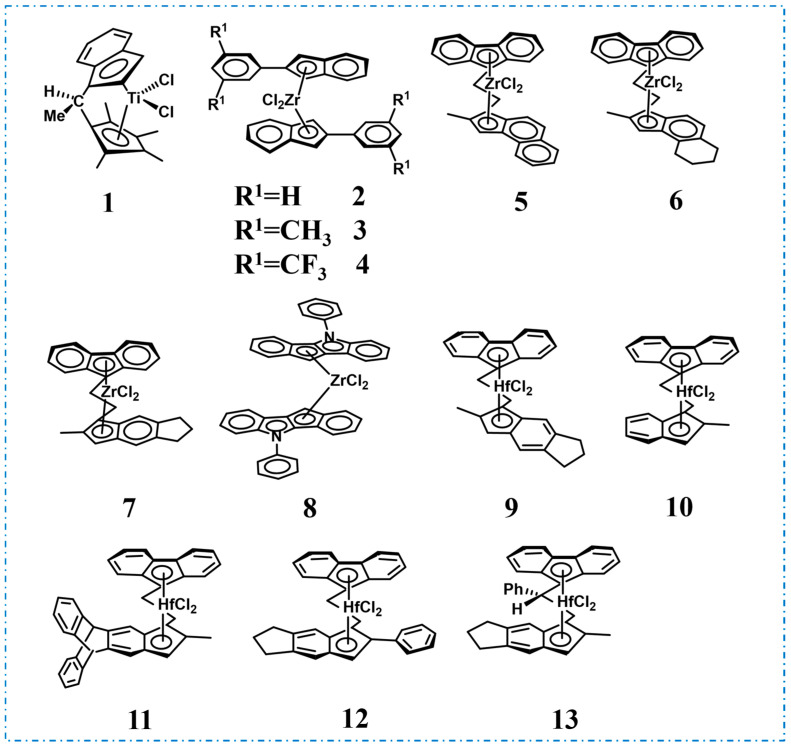
Catalysts for the synthesis of isotactic/atactic stereoblock PBEs.

**Figure 4 polymers-16-02717-f004:**
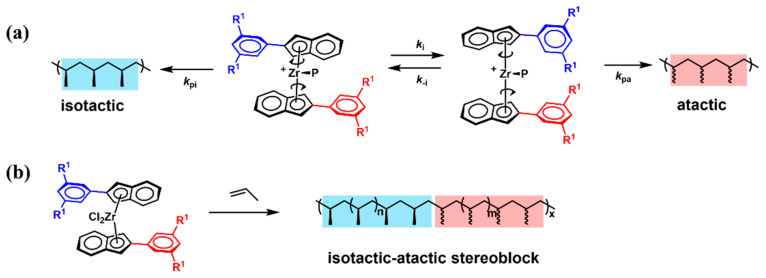
Schematic diagram of oscillating catalyst for the production of stereoblock PBEs.

**Figure 5 polymers-16-02717-f005:**
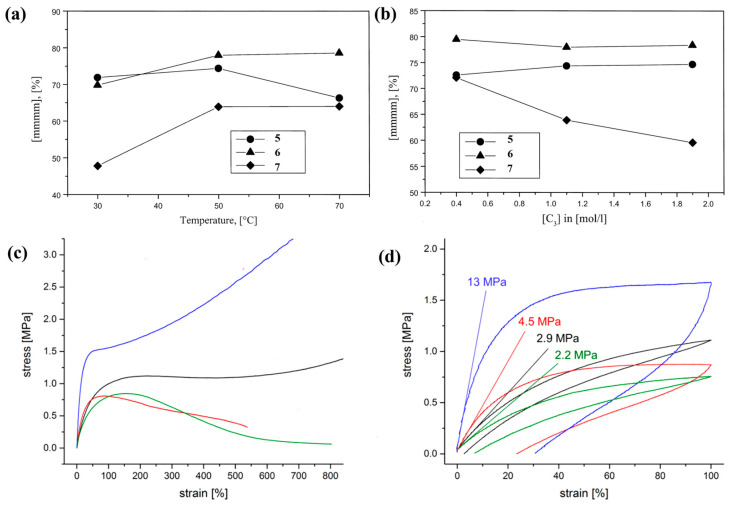
Plot of the propene stereoregularity ([mmmm] pentads) versus (**a**) the polymerization temperature (*T*_p_) and (**b**) the monomer concentration ([*C*_3_]) for catalysts **5**–**7** [36]. Copyright 1999 American Chemical Society. Stress−strain (**c**) and cyclic stress−strain (**d**) hysteresis curves of a compression-molded polypropylene specimen by catalysts **9**–**13** [38]. Copyright 2018 American Chemical Society.

**Figure 6 polymers-16-02717-f006:**
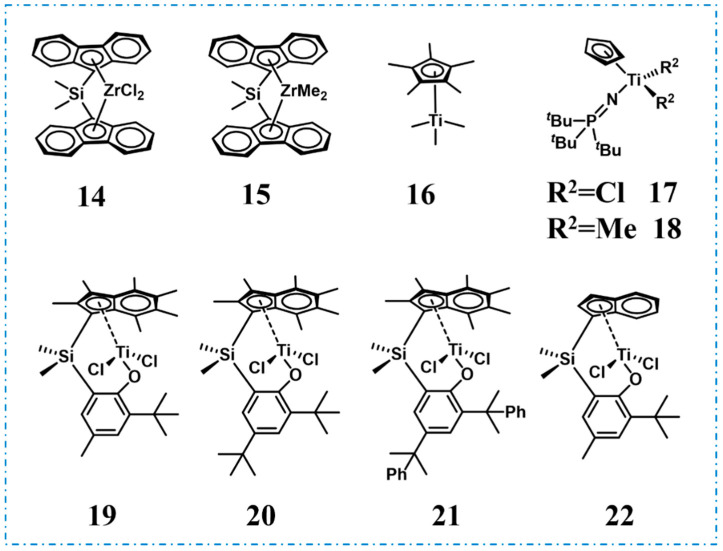
Catalysts for the synthesis of high molecular weight *^a^*PP.

**Figure 7 polymers-16-02717-f007:**
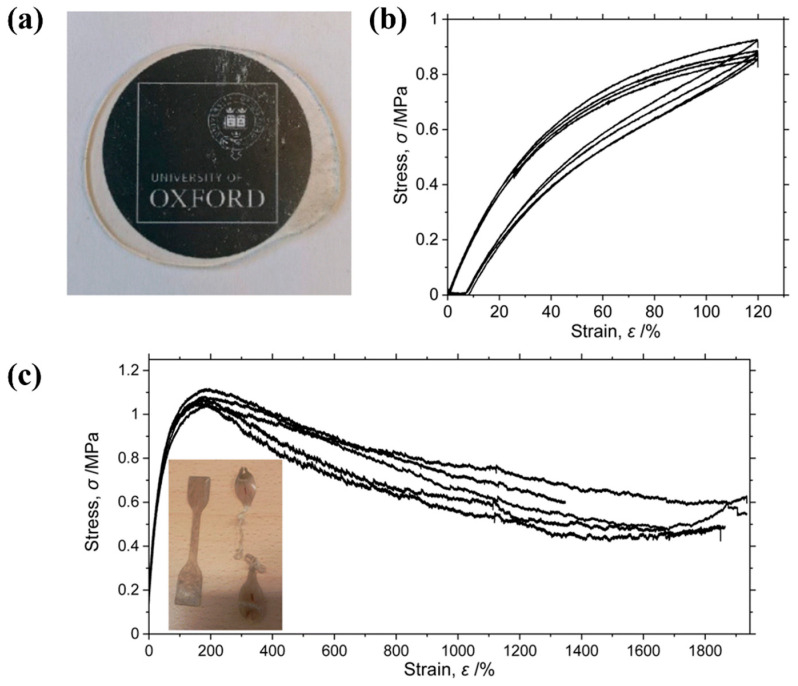
(**a**) Image of a pressed transparent disc of UHMW *^a^*PP. (**b**) Cyclic stretch curves of a UHMW PP sample. The sample was extended to 120% strain cyclically. (**c**) Engineering stress–strain curves (inset: image of dog-bone samples before and after tensile testing) [42]. Copyright 2022 Elsevier B.V.

**Figure 8 polymers-16-02717-f008:**
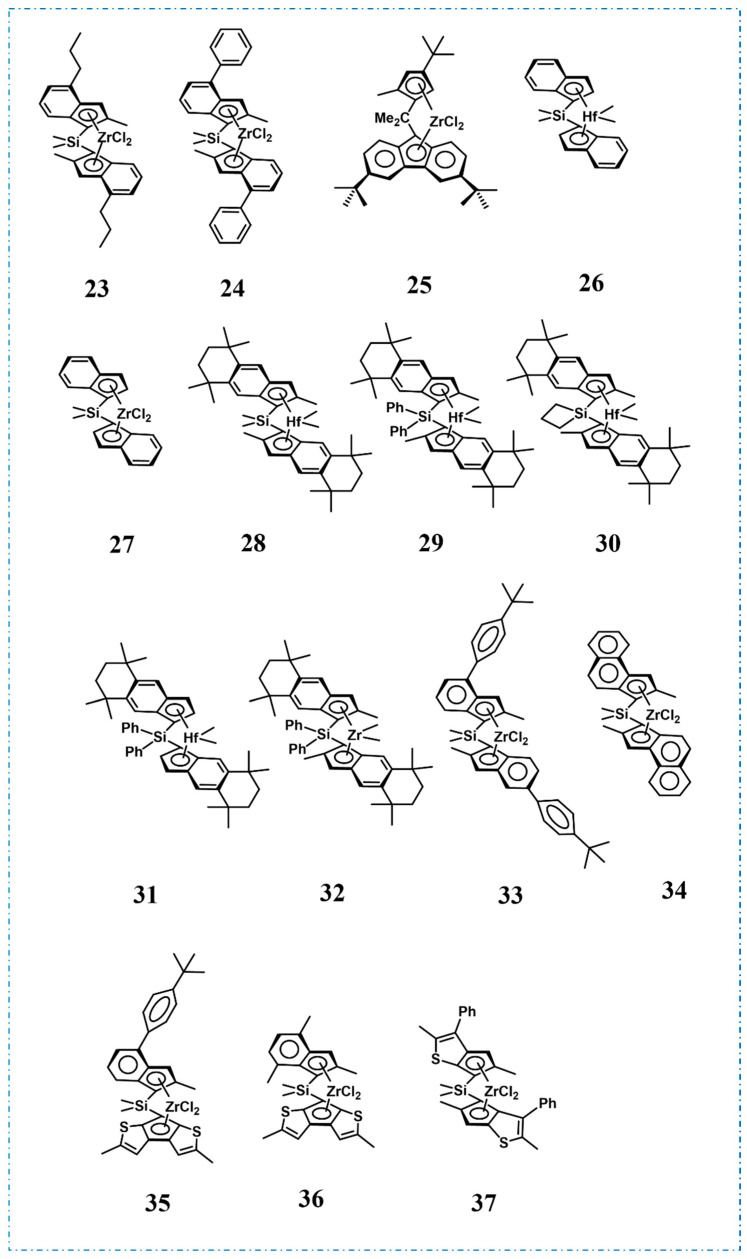
Metallocene catalysts for the synthesis of random copolymer PBEs.

**Figure 9 polymers-16-02717-f009:**
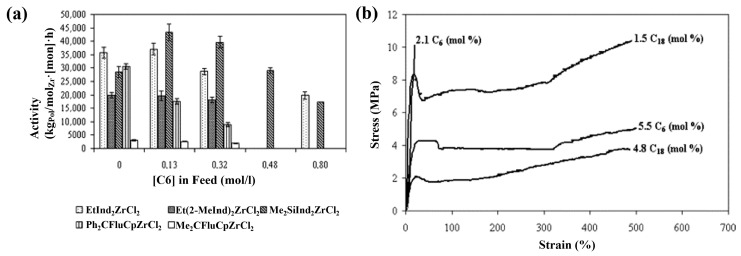
(**a**) The effect of the initial concentration of 1-hexene on activity for the different catalytic systems. (**b**) The stress–strain curves of the copolymers [48]. Copyright 2005 Elsevier B.V.

**Figure 10 polymers-16-02717-f010:**
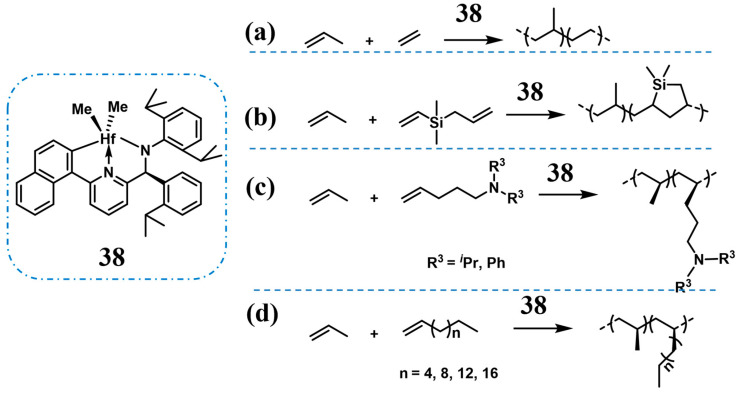
Non-metallocene catalysts for the synthesis of random copolymer PBEs.

**Figure 11 polymers-16-02717-f011:**
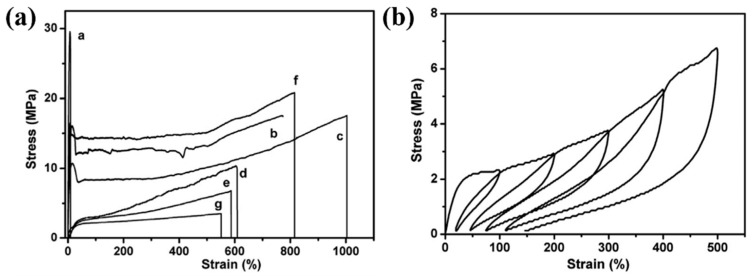
(**a**) Stress–strain curves of the copolymers with different comonomer contents. (**b**) Cycle tensile test with 10.4 mol% comonomer incorporation [54]. Copyright 2015 Elsevier B.V.

**Figure 12 polymers-16-02717-f012:**
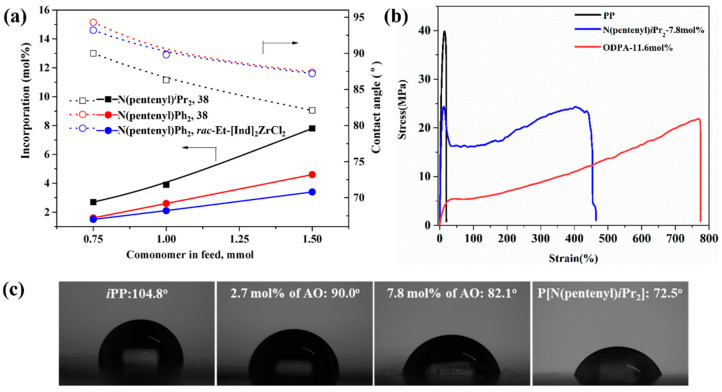
(**a**) Plots of comonomer incorporations and contact angles as a function of comonomer loading varying from 0.75 to 1.5 mmol. (**b**) Comparison of stress-strain curves of the *^i^*PP and copolymers. (**c**) Contact angle of representative (co)polymer samples [55]. Copyright 2019 American Chemical Society.

**Figure 13 polymers-16-02717-f013:**
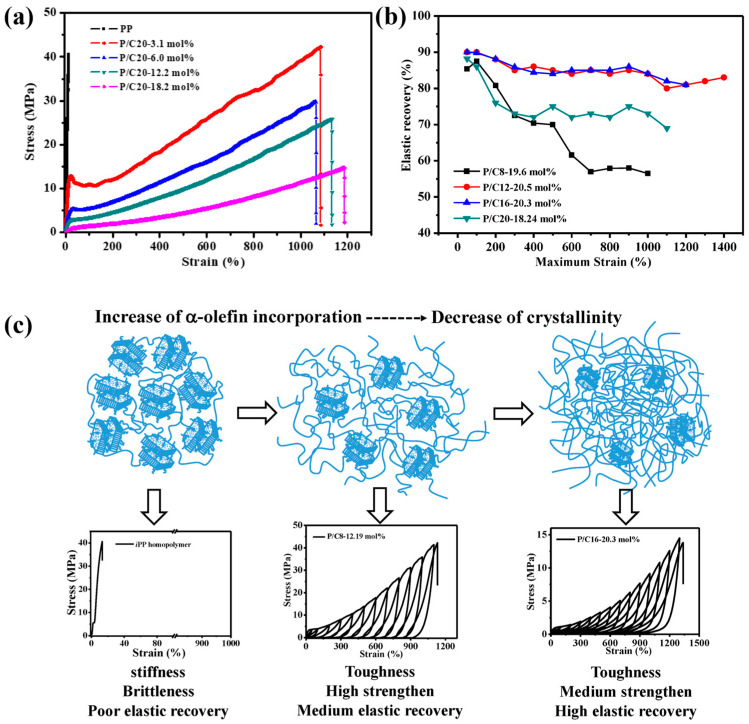
(**a**) The stress–strain curves of P/C20 copolymers with different 1-eicosene incorporation. (**b**) The elastic recovery of copolymers containing 20 mol% comonomers. (**c**) The preparation PBEs through copolymerization of propene and α-olefins [56]. Copyright 2020 MDPI.

**Figure 14 polymers-16-02717-f014:**
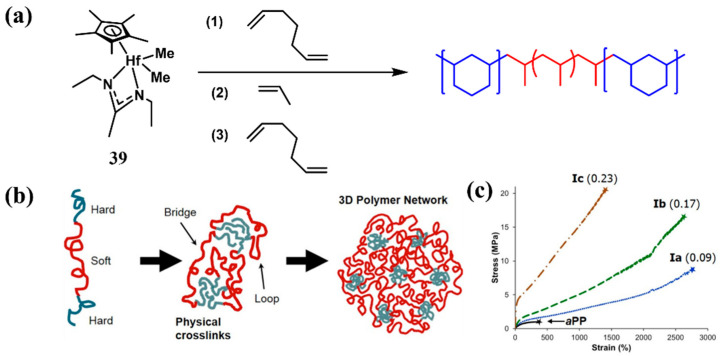
(**a**) Sequential coordination polymerization of HPD, propylene, and HPD. (**b**) Schematic illustration of a triblock TPE. (**c**) Stress-strain curves for a triblock copolymer and *^a^*PP sample of similar molecular weight [61]. Copyright 2015 American Chemical Society.

**Figure 15 polymers-16-02717-f015:**
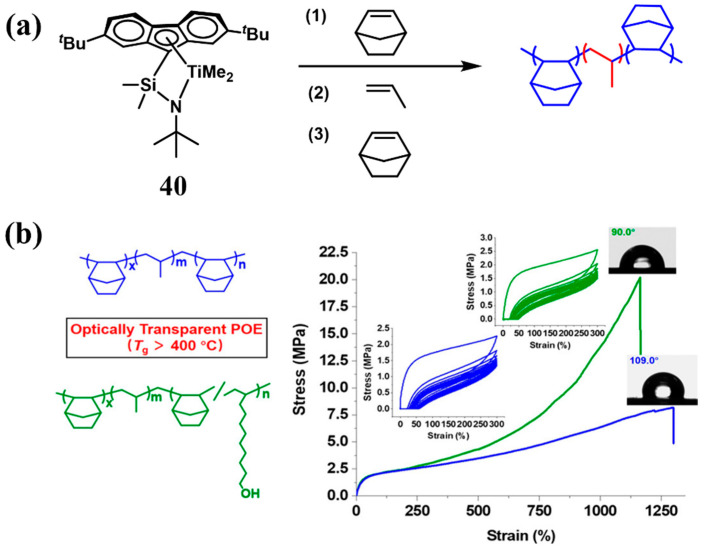
(**a**) Synthesis of the triblock copolymer polynorbornene-*b*-atactic polypropene-*b*-polynorbornene. (**b**) The material properties of triblock copolymers [62]. Copyright 2019 American Chemical Society.

**Figure 16 polymers-16-02717-f016:**
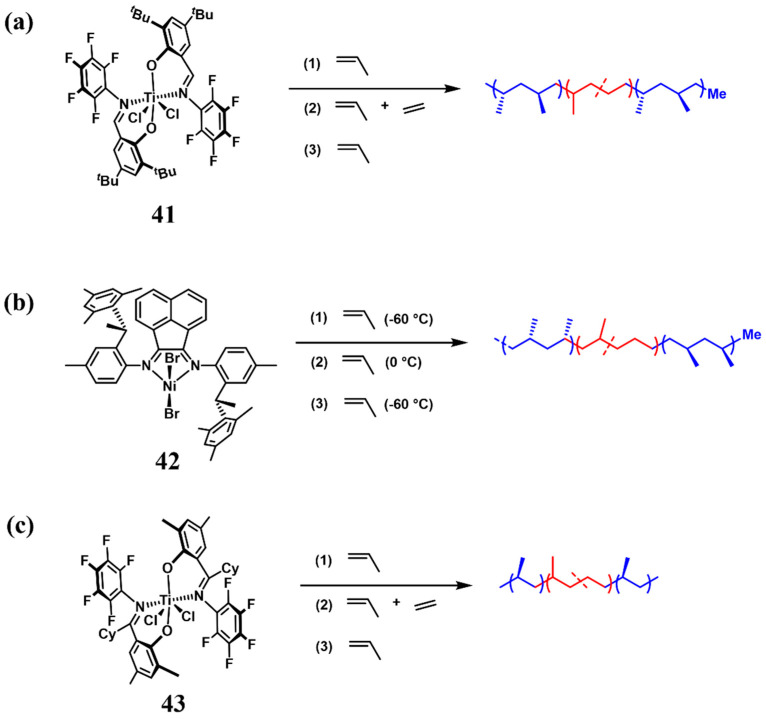
Non-metallocene catalysts for the synthesis of A-B-A triblock copolymer PBEs.

**Figure 17 polymers-16-02717-f017:**
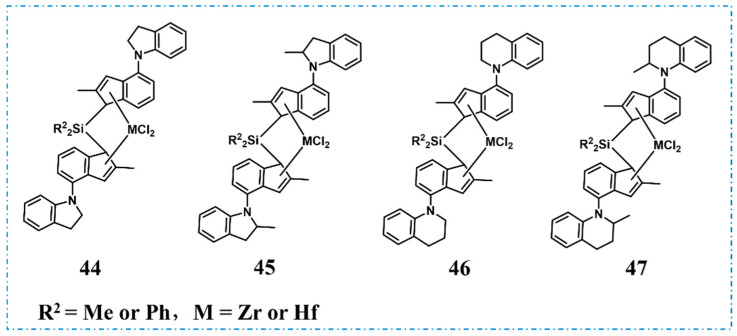
Catalysts for the synthesis of multiblock copolymer PBEs.

## Data Availability

Data derived from public domain resources.

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
