# Peer review of "Recent Advances in Propylene-Based Elastomers Polymerized by Homogeneous Catalysts"

_polymers, 2024, doi:10.3390/polym16192717_

Round 1
Reviewer 1 Report
Comments and Suggestions for Authors
This article reviews the effects of single-site catalysts (metallocene catalysts, constrained geometry catalysts, and non-metallocene catalysts) on the chain structures and properties of PBEs, including homopolymer propylene-based elastomers (hPBEs), random copolymer propylene-based elastomers (rPBEs), and block copolymer propylene-based elastomers (bPBEs). This paper can be considered for publication after it is revised. Some suggestions are given as follows.
1- English of the manuscript needs polishing.
2- More physical interpretation about the presented results can improve the quality of this work.
3- The Abstract in its current form is not sufficient. In particular, it should be supported in a more effective manner by the results obtained during research, because the first part which is read by journal's audience is Abstract and thus it should reflect the main results.
4- A brief discussion about the important parameters affecting the mechanical properties of propylene-based elastomers should be added to the introduction.
5- Conclude the paper with a section on future research directions.
Comments on the Quality of English LanguageEnglish of the manuscript needs polishing.
Author Response
Comment 1: This article reviews the effects of single-site catalysts (metallocene catalysts, constrained geometry catalysts, and non-metallocene catalysts) on the chain structures and properties of PBEs, including homopolymer propylene-based elastomers (hPBEs), random copolymer propylene-based elastomers (rPBEs), and block copolymer propylene-based elastomers (bPBEs). This paper can be considered for publication after it is revised. Some suggestions are given as follows.
Response 1. We greatly appreciate the reviewer’s positive comments and recommendation for publication in Polymers. We carefully considered your following specific comments and made corresponding changes in the revised manuscript.
Comment 2: English of the manuscript needs polishing.
Response 2: We have carefully touched up the manuscript.
Comment 3: More physical interpretation about the presented results can improve the quality of this work.
Response 3: We offered some reasonable explanations for the presented results.
Comments 4: The Abstract in its current form is not sufficient. In particular, it should be supported in a more effective manner by the results obtained during research, because the first part which is read by journal's audience is Abstract and thus it should reflect the main results.
Response 4: We have further detailed the writing logic of this review as well as the specific contents it introduces in the abstract. Following the suggestions, we have incorporated the additions into the abstract.
Comment 5: A brief discussion about the important parameters affecting the mechanical properties of propylene-based elastomers should be added to the introduction.
Response 5: The mechanical properties of PBE are influenced by many parameters, including molecular weight, molecular weight distribution, comonomer type and incorporation content, stereo-regularity and crystallinity. Corresponding description was included into the introduction as suggested.
Comment 6: Conclude the paper with a section on future research directions.
Response 6: For the future research direction of PBEs, we made five recommendations in the Conclusions and outlook, which have been partially supplemented in this revision.
Reviewer 2 Report
Comments and Suggestions for Authors
Dear Editor,
The manuscript (polymers-3170528) entitled ‘‘Recent advances in the propylene-based elastomers by single-site catalysts’’ can be published in the Polymers after major revision. This review article should be more comprehensive and detailed, my suggestions are below.
My suggestions:
1- The closed formulas at the bottom of Figure 7 are not readable, increase the size of the text or change the font, in short, make it readable.
2- 4. Block copolymer propylene-based elastomers, 4. Conclusions and Outlook.
Correct the numbering, Conclusions and outlook should be number 5.
3- Review the abbreviation of journal names, For example,
a-Ref-18: The Chemical Record can be abbreviated as Chem. Rec.
b- Ref-22: In Polymer science: A comprehensive reference can be abbreviated as In Polym. Sci A comprehensive ref.
c- Ref-28: Journal of Macromolecular Science, Part A, Polymer Chemistry should be corrected as Journal of Polymer Science Part A: Polymer Chemistry
And then it can be abbreviated as J. Polym. Sci. Part A: Polym. Chem. 2004, 42, 391-395.
d- Ref-41: Acta Polymerica Sinica can be abbreviated as Acta Polym. Sin.
4- There is some information about single site catalysts at the end of the introduction but not enough. What are the single site catalysts, are there any other than the sandwich type compounds you have given, and mention only one of them, the sandwich type compounds. Single site catalysts are those that contain only one active group per metal and if we consider that this group is the one that initiates the polymerization, your first 12 compounds have two active chlorides per Zr and Hf, which does not fit the definition. Either enrich the description or edit it, especially the title.
5- Your summarization is ok, but you have not drawn any mechanism, it would be more understandable if you draw one general mechanism, which group attacks first, who initiates polymerization, how growth and termination happens.
6- You suggested 47 catalysts, it would be much more understandable if you could summarize it in a table, such as catalyst, temperature, solvent, yield, etc.

Author Response
Comment 1: The manuscript (polymers-3170528) entitled ‘‘Recent advances in the propylene-based elastomers by single-site catalysts’’ can be published in the Polymers after major revision. This review article should be more comprehensive and detailed, my suggestions are below.
Response 1: Thank you very much for your positive recommendation for publication in Polymers! We carefully considered your following specific comments and made corresponding changes in the revised manuscript.
Comment 2: The closed formulas at the bottom of Figure 7 are not readable, increase the size of the text or change the font, in short, make it readable.
Response 2: As suggested, we have made corresponding changes in the revised manuscript.
Comment 3: Block copolymer propylene-based elastomers, 4. Conclusions and Outlook.
Correct the numbering, Conclusions and outlook should be number 5.
Response 3: According to the reviewer’s valuable suggestion, we made corresponding changes in the revised manuscript.
Comment 4: Review the abbreviation of journal names, For example,
a-Ref-18: The Chemical Record can be abbreviated as Chem. Rec.
b- Ref-22: In Polymer science: A comprehensive reference can be abbreviated as In Polym. Sci A comprehensive ref.
c- Ref-28: Journal of Macromolecular Science, Part A, Polymer Chemistry should be corrected as Journal of Polymer Science Part A: Polymer Chemistry
And then it can be abbreviated as J. Polym. Sci. Part A: Polym. Chem. 2004, 42, 391-395.
d- Ref-41: Acta Polymerica Sinica can be abbreviated as Acta Polym. Sin.
Response 4: Thanks for catching such errors for us, we corrected them in the revised manuscript.
Comment 5: There is some information about single site catalysts at the end of the introduction but not enough. What are the single site catalysts, are there any other than the sandwich type compounds you have given, and mention only one of them, the sandwich type compounds. Single site catalysts are those that contain only one active group per metal and if we consider that this group is the one that initiates the polymerization, your first 12 compounds have two active chlorides per Zr and Hf, which does not fit the definition. Either enrich the description or edit it, especially the title.
Response 5: Single-site catalysts are homogeneous catalysts, which can be soluble in hydrocarbon solvents, including metallocene systems, half-sandwich (constrained geometry) titanium catalysts, and non-metallocene systems (Chem. Commun., 2004, 1956-1957, Can. J. Chem. Eng. 2012, 90, 646– 671). In addition to the mentioned sandwich catalysts, there are also some non-sandwich catalysts, such as the catalysts 38, 41-43 we mentioned. In fact, although there are two active chlorides per metal center, each metal center can actually initiate only one polymer chain according to the polymerization mechanism. And we elaborate on the mechanism in the next response.
Comment 6: Your summarization is ok, but you have not drawn any mechanism, it would be more understandable if you draw one general mechanism, which group attacks first, who initiates polymerization, how growth and termination happens.
Response 6: Taking the polymerization of propylene as an example, we describe the polymerization mechanism (Table 2). The catalytically active species is generated by the addition of MAO to the catalyst precursor in most cases and the active species of monocationic alkyl complex is formed. During the process of the polymerization, propylene coordinates as a π-ligand to the unsaturated monocationic alkyl complex. The propylene insertion into the metal alkyl bond is believed to the proceed through a four-member transition state. Then, the process of migratory insertion is continued until termination takes place. The termination occurs predominantly through chain transfer mechanism involving β-H or β-Me elimination. Of course, there are still some other chain transfer processes that we have not described here, such as the transfer to aluminum or monomer. After chain termination reactions, metal hydride or metal alkyl is formed, which is a new catalytically active species, and are able to start a new polymer chain.
Comment 7: You suggested 47 catalysts, it would be much more understandable if you could summarize it in a table, such as catalyst, temperature, solvent, yield, etc.
Response 7: We summarized the data in four tables based on the reaction conditions, and for the data in Table 1 and Table 2, the concentration of propylene is a basis for categorization. Some catalysts were not summarized in the tables due to the lack of relevant data in some of the literature.
Reviewer 3 Report
Comments and Suggestions for Authors
This review is devoted to the analysis of the influence of single-site catalysts on the structure and properties of propylene-based elastomers (PBE). Moreover, the authors examined in detail and thoroughly not only homopolymer hPBE, but also random rPBE and blockcopolymers of propylene with elastomers bPBE. The manuscript clearly presents the structural formulas of various single-site catalysts intended for different types of polymerization. The work turned out to be voluminous and interesting and may be relevant for a wide range of readers involved in polymer synthesis.
However, it is necessary to note a number of issues, the solution of which can make the work better:
1. It is a pity that the names of all the catalysts shown in the figures are not included in the figure captions (or in a separate table). Only the names of individual catalysts are presented in the text.
2. The review does not provide information on the presence/absence of these catalysts in the polymer/copolymer structure. Do they remain at the ends of the chain? How does this affect the structure and properties of polymers?
3. There is no reference to Fig. 5 in the text in lines 140-148. The reference to Fig. 5 appears in the text only on line 225 after Fig. 7.
4. Also, the text first refers to Fig. 8 and only then to Fig. 7.
Author Response
Comments 1: This review is devoted to the analysis of the influence of single-site catalysts on the structure and properties of propylene-based elastomers (PBE). Moreover, the authors examined in detail and thoroughly not only homopolymer hPBE, but also random rPBE and block copolymers of propylene with elastomers bPBE. The manuscript clearly presents the structural formulas of various single-site catalysts intended for different types of polymerization. The work turned out to be voluminous and interesting and may be relevant for a wide range of readers involved in polymer synthesis.
However, it is necessary to note a number of issues, the solution of which can make the work better:
Response 1: Thank you very much for your positive recommendation for publication in Polymers! We carefully considered your following specific comments and made corresponding changes in the revised manuscript.
Comments 2: It is a pity that the names of all the catalysts shown in the figures are not included in the figure captions (or in a separate table). Only the names of individual catalysts are presented in the text.
Response 2: We have given the names of the catalysts in the main text, however, there are a few compounds whose names are not given in the references, so a few of the compounds are not given names.
Comments 3: The review does not provide information on the presence/absence of these catalysts in the polymer/copolymer structure. Do they remain at the ends of the chain? How does this affect the structure and properties of polymers?
Response 3: According to the polymerization mechanism, the catalyst is retained at the end of the polymer chain during the polymerization process, and the steric effect as well as the electronic effect of the catalyst affect the catalytic activity, stereoselectivity, and polymer molecular weight, and so on. However, after the polymerization is completed, the catalyst is removed from the end of the polymer chain, and after the deashing process, the properties of the polymer are almost unaffected.
Comments 4: There is no reference to Fig. 5 in the text in lines 140-148. The reference to Fig. 5 appears in the text only on line 225 after Fig. 7.
Response 4: We changed the number of the images.
Comments 5: Also, the text first refers to Fig. 8 and only then to Fig. 7.
Response 5: We changed the order of images.
Round 2
Reviewer 2 Report
Comments and Suggestions for Authors
Dear Editor,
The manuscript (polymers-3170528) entitled ‘‘Recent advances in the propylene-based elastomers by single-site catalysts’’ can be published in the Polymers after major revision.
My suggestions:
1- My suggestion in my first reading: ‘‘Single site catalysts are those that contain only one active group per metal and if we consider that this group is the one that initiates the polymerization, your first 12 compounds have two active chlorides per Zr and Hf, which does not fit the definition. Either enrich the description or edit it, especially the title’’.
The title of the article should definitely be changed and single-site should be de-emphasized in the sentence, because it is not single-site. Just because someone interpreted it that way, you don't have to write it that way. Does the value of the article decrease when you don't write single-site. If it were a single site, the products would not create so many isomers, especially atactic isomers.
2- iPP and sPP. (Figure 1). Delete the first point.
3- It would be more accurate if the arrow in the mechanism (in Figure 2) was drawn to attack the vinyl group from the bond or Me group rather than from the metal (Mt).
4- In line 375, (Figure 15b, Run 2, Table 4),………………. In Run 2 you say reference 62 in the Table 4, but at the end of the sentence you write reference 63?

Author Response
Comments 1: The manuscript (polymers-3170528) entitled ‘‘Recent advances in the propylene-based elastomers by single-site catalysts’’ can be published in the Polymers after major revision. My suggestions:
Response 1: We greatly appreciate the reviewer’s positive comments and recommendation for publication in Polymers. We carefully considered your following specific comments and made corresponding changes in the revised manuscript.
Comments 2: My suggestion in my first reading: ‘‘Single site catalysts are those that contain only one active group per metal and if we consider that this group is the one that initiates the polymerization, your first 12 compounds have two active chlorides per Zr and Hf, which does not fit the definition. Either enrich the description or edit it, especially the title’’.
The title of the article should definitely be changed and single-site should be de-emphasized in the sentence, because it is not single-site. Just because someone interpreted it that way, you don't have to write it that way. Does the value of the article decrease when you don't write single-site. If it were a single site, the products would not create so many isomers, especially atactic isomers.
Response 2: Based on the reviewer’s valuable suggestion, we replaced the title with Recent advances in the propylene-based elastomers by homogeneous catalysts and also made corresponding changes in the main text.
Comments 3: iPP and sPP. (Figure 1). Delete the first point.
Response 3: We have deleted the first point.
Comments 4: It would be more accurate if the arrow in the mechanism (in Figure 2) was drawn to attack the vinyl group from the bond or Me group rather than from the metal (Mt).
Response 4: We have modified the mechanism.
Comments 5: In line 375, (Figure 15b, Run 2, Table 4),………………. In Run 2 you say reference 62 in the Table 4, but at the end of the sentence you write reference 63?
Response 5: The data of Run 2, Table 4 corresponds to Reference 62 and reference 63 corresponds to a pervious result that as a control result. We also added the reference [62] at the end of Figure 15b, Run 2, Table 4.
Round 3
Reviewer 2 Report
Comments and Suggestions for Authors
Dear Editor,
The manuscript (polymers-3170528) entitled ‘‘Recent advances in the propylene-based elastomers by single-site catalysts’’ can be published in the Polymers as is.
